# Celiac Dietary Adherence Test and Standardized Dietician Evaluation in Assessment of Adherence to a Gluten-Free Diet in Patients with Celiac Disease

**DOI:** 10.3390/nu12082300

**Published:** 2020-07-31

**Authors:** Katarzyna Gładyś, Jolanta Dardzińska, Marek Guzek, Krystian Adrych, Sylwia Małgorzewicz

**Affiliations:** 1Department of Clinical Nutrition, Medical University of Gdansk, Dębinki St. 7, 80-211 Gdańsk, Poland; katarzyna.gladys@gumed.edu.pl (K.G.); sylwia.malgorzewicz@gumed.edu.pl (S.M.); 2Department of Gastroenterology and Hepatology, Medical University of Gdansk, Dębinki St. 7, 80-211 Gdańsk, Poland; marek.guzek@gumed.edu.pl (M.G.); krystian.adrych@gumed.edu.pl (K.A.)

**Keywords:** adherence to a gluten-free diet, refractory celiac disease, dietitian assessment, standardized dietary evaluation

## Abstract

Adherence to a gluten-free diet (GFD) is currently the mainstay of treatment strategy for celiac disease (CD). The aim of our study was measuring a GFD adherence in CD patients using two newly validated methods of dietary assessment—Standardized Dietician Evaluation (SDE) and the Celiac Dietary Adherence Test (CDAT). Ninety-two adults with CD were evaluated by a registered dietitian with extensive experience with the use of SDE and CDAT. Duodenal biopsy was performed and blood was drawn for serum anti-endomysial, anti-deamidated gliadin peptide and anti-tissue transglutaminase antibodies in forty four of those patients. The results of CDAT and SDE were very convergent, but SDE scores better correlated with serologic and histologic findings. As many as 24–52% of study participants did not adhere well enough to a GFD. Insufficient adherence to a GFD in CD patients is still a significant problem. The knowledge about gluten content in food ingredients and additives is very low among adults with CD. SDE is the most accurate method in assessing compliance with a GFD and is especially helpful in determining hidden sources of gluten. The CDAT may be a fast tool for screening for a GFD adherence in CD patients.

## 1. Introduction

Celiac disease (CD) is a lifelong autoimmune disease in which the ingestion of gluten triggers a cellular and humoral immune response leading to small-intestinal mucosal damage in genetically predisposed individuals. It affects approximately 1 in 100 people in Europe and is more common in women than in men [1]. Although new treatment strategies for CD are being developed, until now the only accepted therapy for patients with CD continues to be a lifelong adherence to a gluten-free diet (GFD). It involves the avoidance of foods containing even small amounts of prolamins found in wheat, rye, and barley [2]. Toxicity of avenin, the prolamin of oats, is somewhat disputable [2,3]. Following a strict GFD can induce disease remission and can also reduce the risk of long-term complications such as malignancy (mainly T-cell lymphoma), liver disease, osteoporosis, microcytic anemia, malocclusion, psychiatric diseases, reproductive tract disorders, and other autoimmune diseases [1,2]. However, a significant number of patients do not sustain dietary restriction. According to available data, the compliance with a GFD did not improve over the past twenty years [4,5]. In turn, difficulty adhering to a GFD is associated with reduction in patients’ quality of life and psychological distress [6]. The currently available methods to assess GFD adherence include clinician and dietitian interview, CD-specific titers, or repeated endoscopy with duodenal biopsy. However, it is raised that none of these techniques is sufficiently sensitive. According to the World Gastroenterology Organization (WGO) [7] and The National Institute for Health and Care Excellence (NICE) [8], duodenal biopsy is not the standard way to monitor GFD compliance in CD patients and should be recommended only in patients with persistent or relapsing symptoms without other explanations for those symptoms. It is known that complete mucosal healing in the majority of CD adults may require up to two years or more on GFD, and may sometimes be impossible to achieve despite good adherence to GFD [9,10]. It was also shown that negative serology does not necessarily indicate good GFD adherence and normalization of EMA and tTG antibodies is possible even in patients that consume small amounts of gluten and in whom recovery of the small intestinal mucosa is incomplete [11,12,13,14]. Conversely, mildly elevated tTG antibody levels may also occur in other autoimmune diseases (e.g., Crohn’s disease) and other disorders [15,16]. In experts’ opinion [7,8], serological testing should not be done alone, without detailed evaluation of the clinical situation and a GFD adherence. However, in some groups of patients (especially children, adolescents, and seniors) dietary assessment may be difficult due to unreliable self-reports or a lack of cooperation with a dietitian, therefore, new laboratory tests to monitor adherence are awaited in clinical practice like detection of gluten immunogenic peptides in feces [17,18,19] or intestinal fatty acid binding protein (I–FABP) in the blood [20]. Nevertheless, dietary assessment is still of great importance [7,8]. It is emphasized that such an evaluation should be performed by highly qualified dietitians or physicians with appropriate training, but access to such professionals may be limited. Another important problem is that those assessments are non-objective and thus are not directly comparable, because different methods are used to check a GFD adherence, such as food diaries, 24 h recalls, dietitian interviews, self-reported questionnaires, food frequency questionnaires, or short questions, as it was in the DiGiacomo et al. study [21]. Adherence to a GFD was defined by them as an affirmative answer to the question: “Are you on a gluten-free diet?” [21]. Therefore it seems to be essential to create and introduce into clinical practice and scientific research standardized dietary tools to measure a GFD adherence [22,23]. It would also be desirable that such a tool is straightforward and potentially used even by non-expert personnel [24]. One such method is a fast questionnaire based on four simple questions with a five-level score carried out by Biagi [23,24]. However, this method has some limitations, because not every celiac patient who checks the labels of packaged food can also identify ingredients containing gluten. Two other novel interesting methods of evaluation of a GFD adherence were proposed by Leffler et al. [11,25]. The first is a Standardized Dietician Evaluation (SDE) based on a detailed interview conducted by an experienced dietitian. It is important that daily food diary or 24 h diet recall are only part of this method and not the only form of assessment of compliance with a GFD. The second tool, Celiac Dietary Adherence Test (CDAT) is less time consuming, but according to the authors [11] can easily identify patients at high risk of poor adherence. Both methods were validated in an American population of adults with CD and allowed a reliable assessment of gluten exposure. They can also be helpful in the diagnosis of refractory celiac disease (RCD). Therefore, the aim of the present study was to measure a GFD adherence in Polish CD patients with the use of the CDAT and the SDE and to compare CDAT and SDE scores with results of duodenal biopsies and levels of anti-endomysial (EMA), anti-deamidated gliadin peptide (DGP), and anti-tissue transglutaminase (tTG) antibodies.

## 2. Materials and Methods

The study was conducted from January 2015 to April 2018 and included 92 adults (78 women and 14 men) with CD aged 37.8 ± 12.2 years. All of them were outpatients under the care of the Department of Gastroenterology and Hepatology, Medical University of Gdansk and also members of the Polish Association of People with Celiac Disease.

### 2.1. Selection Procedure

A random selection was performed. Participation in the study was proposed to all patients who came to the gastroenterologist for a follow-up visit and met the criteria for inclusion (age over 18 years and diagnosis of CD based on serological and histological markers, according to British Society of Gastroenterology guidelines for the diagnosis of CD in adults [26]). Patients who had followed a GFD for less than one year were excluded. The decision to perform a control duodenal biopsy among recruited patients was made by the gastroenterologist based on clinical indications according to generally accepted guidelines [7,8].

The protocol was approved by the university bioethics commission (MUG Bioethics Committee approval number is NKBBN/403/201) and informed written consent was obtained from each study participant prior to study enrollment.

### 2.2. Dietitian Assessment

All patients who had given their consent to the study had a personal appointment with a registered dietitian extensively experienced in the dietary management of gastrointestinal disorders and CD. During this 90-min meeting all participants completed a short questionnaire (CDAT) [11], with a subsequent interview (SDE) [11].

### 2.3. Celiac Dietary Adherence Test (CDAT)

The CDAT (created and validated by a panel of experts consisting of gastroenterologists, dieticians, psychologists, and CD patients) takes into account five of the most important aspects of compliance with a GFD: the occurrence of CD symptoms, the patient’s knowledge of the disease and its treatment, confidence in treatment effectiveness, the motivating factors to adherence to a GFD, and self-reported GFD adherence. The questionnaire consists of 7 items on a 5-point Likert scale, and the sum of the numeric values assigned to the answers provides a score ranging from 7 to 35 points. The interpretation was as following: 7 points—excellent GFD adherence; 8–12 points—very good GFD adherence; 13–17 points—insufficient/inadequate GFD adherence, and >17 points—poor GFD adherence. The English version of CDAT was translated into Polish by two independent fluent English-speaking specialists with the forward/backward translation procedure. The tool was then tested out in 12 women and 10 men. All of them found it fully understandable and the twice-obtained results were consistent with the dietary assessment performed by a dietitian.

### 2.4. Standardized Dietician Evaluation (SDE)

The SDE is considered the gold standard when testing compliance with a GFD because it is based on a detailed interview conducted by an experienced dietitian. The SDE consists of three parts:an in-depth analysis of the patient’s diet based on a 24 h or a 3 day nutritional interview;an assessment of the patient’s ability to identify gluten in selected food ingredients and additives (Food Labels Quiz); The Food Label Quiz consists of 28 nutrient components; patient has to determine which food ingredients and additives may contain gluten;Questions mainly about reading the labels on medicines, dietary supplements, or cosmetic products in order to check if they are gluten-free before using them.

The results were recorded by a dietitian on a 6-point Likert scale ranging from 1 to 6, as following: 1 point—perfect GFD adherence; 2 points—good GFD adherence; 3 points—fair GFD adherence; 4 points—poor GFD adherence; 5 points—very poor GFD adherence, and 6 points—no GFD.

### 2.5. Analysis in the Subgroup with Duodenal Biopsy (n = 44)

In 44 patients aged from 18 to 70 years (40.8 ± 12.0 years; BMI 22.4 ± 3.7 kg/m^2^), who had followed GFD for at least year (6.5 ± 7.2 years; median: 3 years), duodenal biopsy was performed and blood was drawn for serum levels of EMA, DGP, and tTG antibodies. DGP and tTG antibodies were assessed using the enzyme-linked immunosorbent assay (Euroimmun, Poland), while EMA antibodies were measured using an indirect immunofluorescence technique (Euroimmun, Poland) in the hospital laboratory. The titers were considered positive according to the manufacturer’s specifications. All studied subjects had total Ig A levels within the reference range. The characteristics of the studied population is presented in Table 1.

### 2.6. Statistical Analysis

Results are expressed as percentages (for categorical variables), mean and standard deviation, or median and interquartile range, as appropriate. The assumption of normality was verified with the Kolmogorov–Smirnov test. A *p*-value < 0.05 was considered to be statistically significant. Comparisons between the two groups were assessed with a Mann–Whitney test or chi-square tests, as appropriate. Additionally, a McNemar’s test was used to assess significant statistical relationships between results of SDE and CDAT assessment. For McNemar analysis, the results of the CDAT and SDE tests were divided into two groups: good or bad. In the case of CDAT, good (Excellent + Very good); bad (Insufficient + Poor) and SDE good (Perfect + Good); and bad (Fair + Poor + Very Poor). Statistical processing of the results was performed with the use of the statistical software STATISTICA PL v 12.0 (Statsoft, Kraków, Poland).

## 3. Results

### 3.1. The Results of CDAT and SDE

The results of CDAT and SDE are presented in Table 2, Figure 1; Figure 2.

Based on the CDAT, 48% of all patients (*n* = 92) presented excellent or very good adherence to a GFD, whereas according to the SDE, 76% of patients presented perfect or good adherence. However, analysis using the McNemar’s test showed a strong correlation between CDAT and SDE scores (χ^2^ = 6.8, *p* = 0.008).

The CDAT and SDE scores did not differ between women and men and in patients with classic and non-classic presentation of the disease.

In the CDAT, patients scored most frequently in questions 1 and 2, which referred to occurrence of symptoms—among 92 participants as many as 84 (91%) admitted that they were bothered by low energy level or by headaches.

### 3.2. Analysis of Results in a Subgroup with the Biopsy

In the subgroup that underwent a duodenal biopsy (*n* = 44) histologic remission (Marsh type 0) was found in 25 out of 44 subjects (56.8%). Among the biopsy group, serologic remission (defined as negative antibody tests) was found, respectively, for 100% of patients with regard to EMA IgG and tTG IgG, 88.6% of patients (39/44) for EMA IgA, 88.1% of patients (37/42) for DPG IgG, 72.7% of patients (32/44) for tTG IgA and 69% of patients (29/42) for DPG IgA.

The patients with and without histologic remission had a similar number of years on GFD, CDAT and SDE scores and antibodies levels except for tTG IgA and DPG IgG (data presented in Table 3).

Significant relationships between the results of the dietitian assessment (SDE and CDAT) and indices of remission in histologic and serologic findings are presented in Table 4.

### 3.3. Other Aspects of Dietary Evaluation (SDE)

Regarding 24 h diet recalls, 25/92 patients (27.2%) consciously or unconsciously consumed food that might have contained gluten. The most common hidden sources of gluten in diet were: “coffee-like” mixtures, tea with added flavor of unknown origin, ham and sausages, fruit yogurts, chocolates, canned fish, and sauces (mayonnaises and ketchups). Only two patients from the studied group (2.2%) ate obvious sources of gluten like wheat tortilla and wheat bread.

In the Food Label Quiz, which is a part of SDE, only 15 patients (16.3%) correctly classified all food ingredients and additives.

## 4. Discussion

Adherence to a GFD is nowadays the main treatment strategy for CD. However a significant portion of adults in our study had suboptimal compliance with the diet, a finding which is in agreement with previously published studies [13,27,28,29,30,31,32]. Most of those studies focused on children or adolescents with CD, while there are still few reports on adults.

The adherence in our study was measured using two standardized methods, so we can compare our results with others who used the same tools. We showed that between 24% (in SDE) and 52% (in CDAT) of patients did not adhere well enough to a GFD. Our results are almost identical to those obtained by Leffler et al. who measured adherence to a GFD with the use of SDE in a group of American patients [11,22]. Importantly, the inclusion and exclusion criteria were the same in our study and Leffler’s. The majority of participants in our study were women, as in Leffler’s study (85% in our study and 77% in Leffler’s). Moreover, the age of the respondents was similar: 37.8 years versus 44.9 years. However, in Leffler’s study the classical form of CD was more frequent (82% vs. 72% in our study), although in the subgroup of our patients with biopsy (*n* = 44) was the same as in Leffler’s study (82%).

Halmos et al. assessed the factors that can influence a GFD adherence in 5310 Australians and New Zealanders with CD using, among other methods, CDAT and found that 61% patients adhered to a GFD, which is similar to our results [33]. Fueyo-Díaz et al. [28], using CDAT in the assessment of a large group of European patients with CD (83.3% women; mean age 35.5 years, 5 years on GFD), reported that only 70% of them adhered perfectly or very well to a GFD, whereas Muhammad et al. [29] stated that only half of UK patients (79% women, mean age 48 years) presented sufficient compliance to a GFD according to CDAT. On the other hand, as many as 86% of Swedish adolescents were adherent to a GFD when CDAT assessment was performed [32]. Those discrepancies can be explained by differences in studied groups’ characteristics (e.g., percentage of concomitant diseases) and organization of health systems among countries, e.g., access to a dietitian and reimbursement of GFD. The latter concerns less wealthy countries such as Poland [34]. But we would like to emphasize that there was a strong correlation between CDAT and SDE scores in our study so our results are similar to those given by Leffler [11].

In our study we analyzed not only serologic titers but also duodenal biopsy results and we observed that SDE scores correlated better than CDAT with other clinical findings. There were significant relationships between the results of SDE and indices of histologic and serologic remission findings, whereas unlike Leffler’s results [11], we did not find any relationship between CDAT score and tTG IgA levels. There are several explanations for this. Firstly, SDE is more detailed and laborious than CDAT. It consists of a 24 h diet recall, food label quiz, and discussion with an experienced dietitian which covers many aspects of a GFD, such as the cost and availability of gluten-free food, eating out, traveling, and socializing with friends. Research highlights that these problems are crucial for patients with CD [35,36]. It is also important that most CD patients can easily identify obviously forbidden products, but recognizing hidden sources of gluten is a major problem. Results of the Food Label Quiz in our study showed that reading and understanding labels is still a valid question in CD patients. Rajpoot et al. also used a food label quiz questionnaire with even more food ingredients than in SDE and pointed out that it is challenging for CD patients to maintain a good adherence to a GFD because of the extensive use of gluten-containing food additives in the food industry [37]. Recently, Paganizza et al. [31] also reported that patients’ knowledge on the gluten content of foods is generally poor and is strongly associated with compliance to a GFD. Jamieson et al. using a food label and ingredient knowledge test showed that 75% of participants made at least one error in identifying gluten-free and gluten-containing foods, which may lead to unintentional gluten consumption and/or unnecessarily restricting safe foods [38]. Therefore WGO and NICE [7,8] highlight the role of the dietitian in monitoring CD treatment and many CD patients notice an urgent need to consult with a dietitian [39,40].

The second important factor in understanding the relationship between SDE and CDAT scores is that SDE focuses strictly on a GFD, and is strongly correlated with small intestinal damage (see Table 4). CDAT additionally assesses general symptoms such as a low energy level or headaches, which can be associated with the presence of concomitant diseases such as hypothyroidism or gastroesophageal reflux. Our results confirmed this observation, because patients who obtained a higher score in the CDAT than in the SDE assessment scored more points for questions on their well-being.

Based on the results of our study, the SDE assessment performed by an experienced dietitian is a very useful measure of a GFD adherence in CD patients. When it comes to CDAT, the process of validating a new tool is recommended in scientific research. In our study, we applied a few principles that made it possible to authenticate the results obtained by us. Obviously, there is a method-related error in the surveys, which is smaller if the tool has been validated for a specific population, so our results should be treated as preliminary. The trial in a bigger multi-center Polish population is needed. However, it seems that the CDAT is a simple, fast, and easily interpretable tool and can be used as the first screening tool for GFD adherence in CD patients. We agree with other authors that CDAT should preferably not be used alone without SDE and biological markers in monitoring of CD treatment [32,41] but there is an increasing amount of data suggesting that under some circumstances CDAT can be used as a single tool in the management of patients with CD [11,29,42,43,44].

There were several limitations in this study. Firstly, our patients were recruited from one academic centre, so the sample was relatively small and consisted mostly of women. We did not perform a power analysis and this could have biased the results. Secondly, more than half of the patients had CD with other autoimmune diseases, which could have affected the CDAT result. Thirdly, both methods can be hampered by the individual’s intentional misreporting of their consumption. Unfortunately, the typical validation process for CDAT has not been carried out, however, in our opinion, the translation process and the simplicity of the test make it possible to apply it in practice. It is worth emphasizing that this is the first study in Poland that shows the benefits of using the CDAT and SDE.

In conclusion, we found that insufficient adherence to a GFD in CD patients is still a significant problem. The knowledge on gluten content in food ingredients and additives is very low among adults with CD. Our study showed that SDE is the most accurate method in assessing compliance to GFD and can help to detect especially hidden sources of gluten. The CDAT may be a fast tool for screening for a GFD adherence in CD patients.

## Figures and Tables

**Figure 1 nutrients-12-02300-f001:**
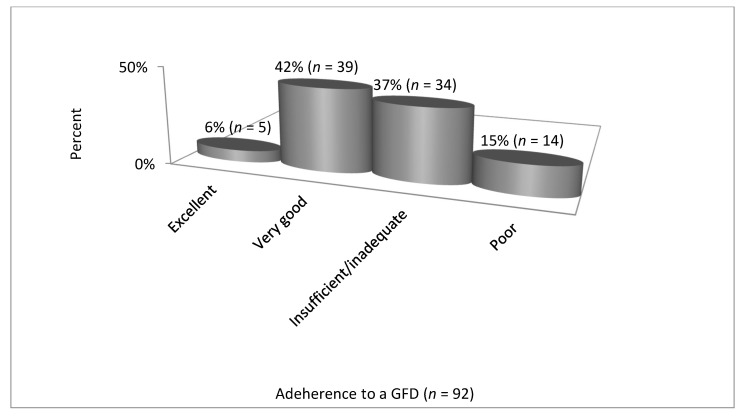
Adherence to a GFD by CD patients—Celiac Dietary Adherence Test, CDAT (*n* = 92). Abbreviations: GDF—gluten-free diet, CD—celiac disease, CDAT—Celiac Dietary Adherence Test.

**Figure 2 nutrients-12-02300-f002:**
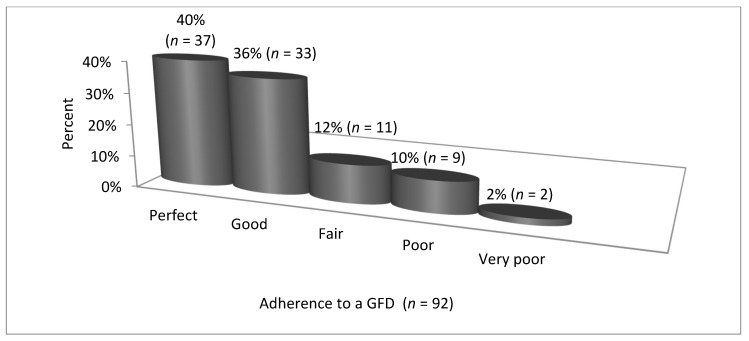
Adherence to a GFD by CD patients—Standardized Dietician Evaluation, SDE (*n* = 92). Abbreviations: GDF—gluten-free diet, CD—celiac disease, SDE—Standardized Dietician Evaluation.

**Table 1 nutrients-12-02300-t001:** Characteristics of the study groups.

Characteristic	All Subjects *N* = 92	Subjects with Biopsy *N* = 44
% Female	84.8	86.4
% Classic presentation	71.8	81.8
% Other autoimmune disorders	53.3	61.4

Differences between the groups were not statistically significant.

**Table 2 nutrients-12-02300-t002:** The results of CDAT and SDE in the study groups (differences between the groups were not statistically significant).

Variable	All Subjects *N* = 92	Subjects with Biopsy *N* = 44
SDE score median (Q1–Q3) Min-max	2 (1–2) 1–5	2 (1–2.5) 1–4
Adherence to a GFD according to SDE *n* (%)	70 (76.1%)	33 (75%)
CDAT score median (Q1–Q3) Min-max	13 (10–16) 7–30	13 (10–16) 7–23
Adherence to a GFD according to CDAT *n* (%)	44 (47.8%)	21 (47.7%)

Abbreviations: CDAT—Celiac Dietary Adherence Test, GDF—gluten-free diet, SDE—Standardized Dietician Evaluation.

**Table 3 nutrients-12-02300-t003:** Results in patients with and without histological remission.

Variable	Remission *n* = 25	No Remission *n* = 19	Remission vs. No Remission *p*
Years on GFD	6.4 ± 6.0	6.7 ± 8.6	0.60
CDAT, points	12.6 ± 3.5	13.8 ± 4.8	0.31
SDE, points	1.7 ± 0.9	2.1 ± 1.1	0.22
tTG IgA, RU/mL	73.4 ± 55.7	103.9 ± 50.0	0.05
EMA IgA	1.6 ± 8.0	61.1 ± 161.0	0.07
DPG IgG, RU/mL	49.6 ± 56.1	96.5 ± 42.8	0.001

Abbreviations: CDAT—Celiac Dietary Adherence Test, DPG IgG—Anti Deamidated Gliadin Peptide Immunoglobulin G, EMA IgA—Anti-Endomysial Antibody Immunoglobulin A, GDF—gluten-free diet, SDE—Standardized Dietician Evaluation, tTg-IgA—Tissue Transglutaminase Immunoglobulin A.

**Table 4 nutrients-12-02300-t004:** Relationships between the results of the dietitian assessment (SDE and CDAT) and indices of remission in histologic and serologic findings.

Index	SDE	CDAT
Histological remission	χ^2^ = 6.52; *p* = 0.010	χ^2^ = 0.04; *p* = 0.801
Absence of tTG IgA	χ^2^ = 12.91; *p* = 0.003	χ^2^ = 2.37; *p* = 0.123
Absence of EMA Ig A	χ^2^ = 22.70; *p* = 0.000	χ^2^ = 10.21; *p* = 0.010
Absence of DPG IgA	χ^2^ = 11.21; *p* = 0.000	χ^2^ = 1.56; *p* = 0.210

Abbreviations: CDAT—Celiac Dietary Adherence Test, DPG IgA—Anti Deamidated Gliadin Peptide Immunoglobulin A, EMA IgA—Anti-Endomysial Antibody Immunoglobulin A, SDE—Standardized Dietician Evaluation, tTg-IgA—Tissue Transglutaminase Immunoglobulin A.

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
