# Peer review of "Celiac Dietary Adherence Test and Standardized Dietician Evaluation in Assessment of Adherence to a Gluten-Free Diet in Patients with Celiac Disease"

_nutrients, 2020, doi:10.3390/nu12082300_

Round 1

Reviewer 1 Report

In this manuscript, the authors report the implementation of the Celiac Dietary Adherence Test (CDAT) and Standardized Dietitian Evaluation (SDE) methods for assessment of compliance to a gluten free diet in a Polish population. These two methods have already been developed and validated by other colleagues. The authors implemented these methods in a cohort of subjects and investigated any associations between the test results, study subject characteristics and disease remission. The authors successfully applied both methods in their cohort and reported the findings in detail, which led to adequate conclusions, in which the SDE method appears to be more efficient in assessing compliance to a GFD. The authors reiterate the importance of SDE, and raise caution on using the CDAT method by itself. Although the findings are well reported and the manuscript evidences implementation of these methods in a different population, some points should be addressed .

The two methods were developed by colleagues in the United States and implemented in an American population. The authors acknowledged this. The manuscript by Leffler is cited in the discussion. Given that this manuscript focuses on the same methods in a different population, it is advised that the authors discuss in more detail differences and similarities between their study and Leffler’s. The authors mentioned that “there was a strong correlation between CDAT and SDE scores in our study so we can confirm results given by Leffler”. It is unclear how the authors can confirm results. The findings can be similar, however, similar findings cannot “confirm” another set. Perhaps the wording should be directed more to reproducibility, or that this indicates that both populations are similar. Therefore, it is important that both populations from both studies is compared and mentioned in the discussion. This will enrich the paper.

Please spell out acronyms in the figure legends. Also please label y axes in both figures. If possible, please indicate also the number of subjects in each bar, since a percentage is a relative measurement and sometimes can be misleading.

Reviewer 2 Report

The manuscript entitled “Celiac Dietary Adherence Test and Standardized Dietician Evaluation in assessment of adherence to a gluten-free diet in patients with celiac disease” presents an interesting issue, and may be interesting for readers, however it requires some amendments.  

ABSTRACT:

  • The abstract should follow the style of structured abstracts, but WITHOUT headings.
  • Scientific writing has traditionally been third person, passive voice – author should avoid first person.

INTRODUCTION:

  • In this session Authors presented the information associated with the CD, but little information is presented associated with applied test. This section should be briefly presented – what do we know and what is the background for this study. Some detailed information about other studies are necessary. The good background should present the history of problem, the current knowledge and scientific "gap", and then authors should present how their study could fill this gap to justify the study.

MATERIALS AND METHODS:

  • Line 54 – it should be dots instead of comas (37.8±12.2)
  • The power of the study should be presented (due to the fact, that the number of men is quite small). It is well know that this specific group of patients is quite demanding, however this factor should be presented (and could be discussed in the limitations section as a bias).
  • Please add the inclusion and exclusion criteria to the study
  • Were the questionnaires (CDAT; SDE) previously validated? What were the accuracy and consistency of this questionnaire? Please add detailed information of Polish version of this questionnaires. This information is crucial for validity of the results.
  • In order to use the questionnaire in different then original language, the questionnaire should be was translated (sometimes transculturally adapted) by 2 independent fluent English-speaking specialists, with the verification of meaning and sense of sentences and necessary adaptation. The back-translation, consensus in the group of specialists (including translator, nutritionist, dietitian, nutritional educator), pre-testing, testing in a sample of respondents, verification of answers, should be also performed.
  • In lines 79; 81; 84; - please add the consecutive number of 3 parts.
  • Why authors used “Kolmogorov–Smirnov test” instead of “Shapiro–Wilk test” – please justify and present the reason of using it (in response letter, not in the manuscript)
  • the criteria for remission should be presented

DISCUSSION:

  • Figure 1 and 2 are redundant
  • This section must be improved. Authors should relate the findings to those of similar studies and point the differences and similarities between the studies. Authors should add the appropriate references including those which referred the Polish population of CD (if available).
  • At the end of the discussion the limitation section should be presented (including the small number of patients; response bias; etc.)

Other comments:

  • Some minor problems with style (e.g., unnecessary bolding in legend of table no 4)

Reviewer 3 Report

30 – Coeliac Disease – spelling differs from Line 14

35 – Prolamines – For consistency, use “prolamin” and “prolamins” not “prolamines”

40 – Reference needed at end of statement on diseases.

50-51 – abbreviations should be spelled out the first time they are used

74 – the test cannot have < 7 points, as the scale is from 7-35 points (according to line 73)

121-122 - McNemar’s test are only used for 2x2 conditions (Yes vs. No under two different treatment conditions, for example).  In this case, lumping  Excellent + Very Good (CDAT) together and comparing it to Perfect + Good (SDE) is somewhat arbitrary, as the Likert scales used in these measures are ordinal, not interval, and the groupings of different values into these named categories is not necessarily comparing apples to apples.  Likely this was done in consideration of Leffler (2008), who binned SDE results using Perfect + Good in their original CDAT validation study , but it should be stated by the authors that this is why they have chosen to bin results in this manner (if in fact it is the reason) to fit it to a McNemar’s test.

128-128 – Please capitalize the abbreviation  “EMA” (as was done elsewhere in the MS)

166 – Capitalize abbreviation EMA instead of “EmA”

In general, there is no need to repeat abbreviations in the figure/table legends, which have been already given elsewhere.

Reviewer 4 Report

Brief summary

It is an article that values the importance of other methods in the follow-up of Celiac Disease, because it is known that adherence to the diet is not easy and that dietary transgressions are frequent. It is necessary to make modifications in each part of the article.

Broad comments

ABSTRACT: the objectives do not correspond to those of the text. The objective is stated to be: “we aimed at measuring a GFD adherence in CD patients using two newly validated methods of dietary assessment”. The objectives of the work are not clear in the text. The conclusions respond to other objectives. It would be necessary to specify all the abbreviations and to homogenize numbers and letters (44, 92).

BACKGROUND: It adequately explains the disease before moving on to the problem in question. There is no adequate bibliographic review to introduce the topic: current problems are not specified and why this study has been done is not justified.

The objectives set out in the introduction are different from those of the summary, it is stated: “the aim of the present study was to adapt an American version of the CDAT and the SDE for Polish patients with CD and to compare CDAT and SDE scores with results of duodenal biopsies and levels of EMA, DGP and tTG antibodies. ” I think it should be specified the aim of the paper: if it is  to describe the adherence to DSG of a cohort, it should be explained this way, introducing previously with a better review of the literature, which includes problems of non-adherence or other studies in the that there is a lack of adherence. If what is intended is to carry out an evaluation study of two diagnostic tests comparing to the gold standard, it must explain why this is done. Furthermore, nothing is mentioned in other articles that have already compared different diagnostic tests (Gerasimidis 2018). Silvester 2017, Biagi 2009, Biagi 2012, Husby 2019, and other articles on gluten peptide in feces should also be incorporated (Comino 2015).

METHODS: In my opinion, Table 1 needs to be completely changed. Percentages are randomly compared, but not specified. What autoimmune diseases? What do they call classic presentation? Why is a data from the questionnaires incorporated in Table 1, which is also supposed to define the baseline characteristics of the population? It would be necessary to make a table 1 with the basal characteristics of the population and present it in results. If the study aims to measure the adherence of the cohort, you must specify whether there has been previous work on the questionnaire, adapted to the population that performs the questionnaires, for validation. If what you want is to verify if these two tests are useful (diagnostic test evaluation study), it is necessary to clearly define the methodology, define the gold standard, comparison, etc. and perform positive and negative probability ratios. In a study to evaluate diagnostic tests, the odds ratios should be included, that is, the change from the pre-test probability to the post-test probability, the percentage of patients who would benefit from the test result in the form of diagnostic and / or therapeutic decisions. Confidence intervals are not included.

RESULTS: Instead of table 4, a table with the numerical data should be explained. There are no statistically significant differences between patients in remission and those who are not, which in my opinion is the most important.

DISCUSSION: The first part is expressed as an introduction, it should go in the introduction instead of the discussion. The most important result for researchers (first line) does not appear in results, only in graphs. The limitations of the study and how they can affect the conclusions are not adequately discussed. The conclusions do not respond to the objectives of the work.

References suggested:

  • Biagi, F., Andrealli, A., Bianchi, P. I., Marchese, A., Klersy, C., & Corazza, G. R. (2009). A gluten-free diet score to evaluate dietary compliance in patients with coeliac disease. The British journal of nutrition, 102(6), 882–887. 
  • Biagi, F., Bianchi, P. I., Marchese, A., Trotta, L., Vattiato, C., Balduzzi, D., Brusco, G., Andrealli, A., Cisarò, F., Astegiano, M., Pellegrino, S., Magazzù, G., Klersy, C., & Corazza, G. R. (2012). A score that verifies adherence to a gluten-free diet: a cross-sectional, multicentre validation in real clinical life. The British journal of nutrition, 108(10), 1884–1888. 
  • Husby, S., Murray, J. A., & Katzka, D. A. (2019). AGA Clinical Practice Update on Diagnosis and Monitoring of Celiac Disease-Changing Utility of Serology and Histologic Measures: Expert Review. Gastroenterology, 156(4), 885–889. 
  • Barratt SM, Leeds JS, Sanders DS. Quality of life in coeliac disease is determined by perceived degree of difficulty adhering to a gluten-free diet, not the level of dietary adherence ultimately achieved. J Gastrointestin Liver Dis. 2011;20:241-5.
  • Comino I, Fernández-Banares F, Esteve M, Ortigosa L, Castillejo G, et al. Fecal gluten peptides reveal limitations of serological tests and food questionnaires for monito­ring gluten-free diet in celiac disease patients. Am J Gas­troenterol. 2016;111:1456-65.
  • Silvester JA, Kurada S, Szwajcer A, Kelly CP, Leffler DA, et al. Tests for serum transglutaminase and endomysial antibodies do not detect most patients with celiac disea­se and persistent villous atrophy on gluten-free diets: a meta-analysis. Gastroenterology. 2017;153:689-701.
  • Comino I, Real A, Vivas S, Siglez MA, Caminero A, et al. Monitoring of gluten-free diet compliance in celiac pa­tients by assessment of gliadin 33-mer equivalent epito­pes in feces. Am J Clin Nutr. 2012;95:670-7.
  • Gerasimidis K, Zafeiropoulou K, Mackinder M, Ijaz UZ, Duncan H, Buchanan E, et al. Comparison of clinical methods with the faecal gluten immunogenic peptide to assess gluten intake in coeliac disease. J Pediatr Gastroen­terol Nutr. 2018;67:356-60.

Specific comments

Line 36: “toxicity…” this sentence must be cited.

Line 39-40:  “following…” this sentence must be cited.

Line 40-41: “Experts”… this sentence must be cited.

Line 45: in my opinion a test that was done in 2007 is not recently.

Line 48: objectives must be clarified.

Line 61-62: this sentence is repeated.

Line 78: “is considered the gold standard…” this sentence must be cited.

Line 90-91: must be unified (40.8 ± 12.0 years; BMI 22.4 ± 3.7 kg/m2), who had followed GFD for at least 1 year (mean ± SD: 6.5 ± 7.2 years), here maybe could be interest the use of median value.

Line 155-174: it must be modified, it is more an introduction than a discussion text.

Line 176:  “our results…” is repeated.

Line 189: “There were…” must be clarified, because there is no significance results between the group with histolical remission and the group without remission in SDE mean of questionary.  

Line 205: “our results..” is not shown into the text.

Round 2

Reviewer 2 Report

Authors have made a great effort to improve the manuscript, however some corrections and responses to comments are required.

Minor comments:

  • Due to the fact, that authors did not perform a power analysis, it must be indicated as a limitation of the study in the limitations section (at the end of the discussion).
  • The justification for exclusion of patients who had followed a GFD for less than one year should be presented (e.g. please add the references or present medical justification of this specific period)
  • As authors stated in response “the Shapiro-Wilk test is recommended for small groups”. As in the presented manuscript authors have small group (less than 100 patients), that was my question. After this explanation I appreciate statistical knowledge of authors, however in further analysis Shapiro-Wilk test for such heterogeneous data will be more suitable. The differences in analysis may be not big, but they may appear
  • If authors believed that Figure 1 and 2 are important - please improve it. In the present form it looks like as in a student work.
  • In table 3 – there are some typos (inconsistencies associated with the space before/ after symbol ±).

Major comment:

However my major comment is associated with the lack of validation of the Polish version of the tools. The authors should conduct at least standard forward- and back-translation procedure. Errors in the forward- or back-translation processes should be corrected and the process should be repeated. At least repeatability in a period of approximately 2-3 weeks should be conducted. In the present form of the manuscript there is a doubt of adequate reliability and validity of the tools. This issue must be discussed deeply in the discussion section – how it could influence the results, as well as must be presented in a limitation section.

Reviewer 4 Report

Changes made in the text have significantly improved the document, especially the background and the discussion, however there are some questions that need to be answered:

- Authors have used a not yet validated tool for Polish population, so a validation procedurre must be done so that conclusions could be interpreted truly. Authors defends that is presented the results of a pilot study but I think it is no enough and vailidity cuould be compromissed. This issue must be discussed in the discussion and authors have to explain how this could affect results of the study.
